# Biological Effects of HDAC Inhibitors Vary with Zinc Binding Group: Differential Effects on Zinc Bioavailability, ROS Production, and R175H p53 Mutant Protein Reactivation

**DOI:** 10.3390/biom13111588

**Published:** 2023-10-28

**Authors:** Brianna M. Flores, Chandana K. Uppalapati, Agnes S. Pascual, Alan Vong, Margaux A. Baatz, Alisha M. Harrison, Kathryn J. Leyva, Elizabeth E. Hull

**Affiliations:** 1Biomedical Sciences Program, College of Graduate Studies, Midwestern University, 19555 N 59th Avenue, Glendale, AZ 85308, USA; brianna.flores@midwestern.edu (B.M.F.); apascu@midwestern.edu (A.S.P.); margaux.baatz@midwestern.edu (M.A.B.);; 2Arizona College of Osteopathic Medicine, Midwestern University, 19555 N 59th Avenue, Glendale, AZ 85308, USA; 3Department of Microbiology & Immunology, College of Graduate Studies, Midwestern University, 19555 N 59th Avenue, Glendale, AZ 85308, USA; cuppal@midwestern.edu (C.K.U.); kleyva@midwestern.edu (K.J.L.)

**Keywords:** HDAC inhibitors, zinc bioavailability, p53 reactivation, R175H mutant p53, ROS production

## Abstract

The coordination of zinc by histone deacetylase inhibitors (HDACi), altering the bioavailability of zinc to histone deacetylases (HDACs), is key to HDAC enzyme inhibition. However, the ability of zinc binding groups (ZBGs) to alter intracellular free Zn^+2^ levels, which may have far-reaching effects, has not been explored. Using two HDACis with different ZBGs, we documented shifts in intracellular free Zn^+2^ concentrations that correlate with subsequent ROS production. Next, we assayed refolding and reactivation of the R175H mutant p53 protein in vitro to provide greater biological context as the activity of this mutant depends on cellular zinc concentration. The data presented demonstrates the differential activity of HDACi in promoting R175H response element (RE) binding. After cells are treated with HDACi, there are differences in R175H mutant p53 refolding and reactivation, which may be related to treatments. Collectively, we show that HDACis with distinct ZBGs differentially impact the intracellular free Zn^+2^ concentration, ROS levels, and activity of R175H; therefore, HDACis may have significant activity independent of their ability to alter acetylation levels. Our results suggest a framework for reevaluating the role of zinc in the variable or off-target effects of HDACi, suggesting that the ZBGs of HDAC inhibitors may provide bioavailable zinc without the toxicity associated with zinc metallochaperones such as ZMC1.

## 1. Introduction

Interest in harnessing the therapeutic potential of histone deacetylase inhibitors (HDACi) continues despite the many obstacles encountered in moving these compounds into the clinic including the limited effectiveness on solid tumors, development of resistance when used as monotherapy, and mild to severe adverse effects observed (reviewed in [1]). As their name suggests, inhibitors that target class I histone deacetylases alter the acetylation state of histones to promote transcriptionally active chromatin structure and alter gene expression. However, the resulting histone hyperacetylation leads to non-specific and broad-ranging changes in gene expression, which limits the utility of HDACi. Additionally, HDAC enzymes deacetylate many non-histone proteins and the inhibition of HDAC activity contributes to wide-ranging shifts in the activity of multiple signaling pathways. Of particular relevance is that HDAC inhibition leads to the addition of activating acetyl groups on p53 [2]. Specifically, HDACi have been shown to acetylate p53 to promote DNA binding, transcriptional activation, and protein stability [3,4,5,6]. Furthermore, the fact that HDAC enzymes do not function independently but form several multi-protein complexes [7,8] adds an additional layer of complexity that needs to be taken into account when considering HDACi therapy. Thus, a deeper understanding of the multiple levels of HDACi action is needed to improve the utility of these potent therapeutics.

Although the in vivo activity of these compounds is much more complex than expected, HDACi have been well-characterized using purified HDAC enzymes and in vitro assays. Structural studies have defined distinct activities for each of the three common regions: the cap binds to the surface of the enzyme while the linker allows positioning of the zinc binding group (ZBG) so that it can interact with a zinc ion, which is required for HDAC enzyme activity. The binding of Zn^+2^ to the ZBG is crucial to the inhibitory activity of HDAC inhibitors (reviewed in [9]). The premise of this work is that the HDACi ZBG contributes to the complexity of HDACi in vivo due to its ability to alter zinc availability and affect the activity of zinc-requiring enzymes. 

Free Zn^+2^ has the potential to initiate the production of reactive oxygen species (ROS). The potential for cellular damage downstream of elevations in zinc is seen in the necessity of multiple homeostatic mechanisms operating to maintain low intracellular free Zn^+2^ concentrations. Zinc is essential for the activity of many DNA binding proteins through motifs such as the zinc finger, and most cellular zinc is sequestered in this bound pool. As it is a redox-active metal that can generate ROS via the Fenton reaction, levels of free Zn^+2^ are carefully controlled and maintained at picomolar ranges depending on the measurement [10,11,12]. Total cellular zinc is ~6 orders of magnitude higher at ~200–300 µM. Zinc binding proteins represent approximately 10% of the genome, and 25% of transcription factors, and zinc binding regulates the function of these proteins [13], suggesting that alterations in free Zn^+2^ may have far-reaching effects. Thus, broadening the understanding of HDACi as modulators of zinc bioavailability may contribute to understanding the activities of HDACi.

HDACi utilize a variety of ZBGs to inhibit HDAC activity. In this work, we compare two class I HDACi that have different zinc binding domains (Appendix A, reviewed in [9,14]). MS275 (Entinostat) is a benzaminde HDACi that predominantly inhibits HDAC1 and HDAC3, and FK228 (Romidepsin) is a cyclic peptide that principally inhibits HDAC1 and HDAC2, respectively, in in vitro assays. Thus, although both compounds target class I HDACs, the ZBGs of each are fundamentally different. Specifically, the cysteine containing cyclic peptide FK228 only binds Zn^+2^ after reduction in the cytosol. In contrast, the structure of the benzamide group of MS275 does not change, and due to the concentration gradient between extracellular and intracellular zinc, it is expected to carry bound Zn^+2^ into the cell. Thus, we hypothesize that each ZBG is expected to have opposing effects on intracellular Zn^+2^ and that any effect will be transient due to the tight homeostasis associated maintenance of a low intracellular free zinc concentration. However, transient increases in Zn^+2^ have been shown to be sufficient to reactivate the R175H mutant p53 protein [15]. 

This work utilizes the R175H mutant p53 as an exemplar of a protein that requires zinc for wild-type structure and function for two reasons. First, the zinc binding of this mutant protein has been extensively studied due to the intense interest in the reactivation of this conformational mutant. The substitution of the arginine with histidine at position 175 disrupts the coordination of the essential zinc atom by C176, H179, C238, and C242. Consequently, the R175H mutant protein binds Zn^+2^ with a Kd of 2 nM, twice that of the WT protein, and above the concentration of free Zn^+2^ in the cell [16,17,18]. Second, this conformational switch from misfolded to WT conformation provides a ready biological assay for Zn^+2^ levels. Studies aimed at the restoration of function to the R175H GOF p53 mutant protein led to the identification of ZMC1 (first identified as NSC 319726). This compound functions as a zinc ionophore to restore activity to the R175H GOF p53 mutant protein, but generates significant ROS due to the chelation of copper, which limits the use of ZMC1 in some but not all combinatorial therapeutic approaches [17,19,20]. 

Mutations in *p53* occur in approximately half of cancers, and approximately 70% of these are missense mutations in the DNA binding domain. As this type of *p53* mutation is both common and confers pro-oncogenic properties, these gain-of-function (GOF) p53 mutant proteins have been extensively characterized. Therapeutic strategies to restore wild-type (WT) function to these proteins (reviewed in [18]) hold tremendous promise. Although it has proven difficult to restore WT function to GOF mutant p53 proteins in which mutation results in the alteration of an amino acid in the region that directly binds to the DNA response element, those GOF mutant p53 proteins with altered structure (or conformational mutants) may present a more tractable problem. Thus, this work focused on the ability of HDACi to restore function to the R175H GOF mutant p53 protein for two reasons. First, the activity of this mutant protein depends on Zn^+2^ binding, which has been well-characterized, thus establishes a framework for the interpretation of zinc levels. Second, this particular GOF p53 occurs frequently in tumors, and the restoration of WT function to the R175H GOF mutant p53 protein is of considerable interest [17,18,19,21,22,23]. 

The data presented combines several cell culture and in vitro experimental approaches to segregate the effects of ZBGs, zinc concentration, and R175H reactivation. Specifically, to address the ability of the ZBG to alter Zn^+2^ levels, changes in free Zn^+2^ concentration immediately following HDACi treatment were measured. To directly assay the ability of the ZBG to alter bioavailable zinc, the ability HDACi to affect refolding of R175H was measured in an in vitro p53 RE binding assay. Cell culture experiments probed the effect of the different ZBGs on R175H reactivation when the complexity of HDACi treatments must be considered. Taken together, the data presented here suggest that differences in the ZBG of HDACi should be considered when evaluating the effects of this class of therapeutics. 

## 2. Materials and Methods

### 2.1. Cell Lines, Extracts, Treatments, and Culture Conditions

Nuclear extracts obtained from the SKBr3 human breast cancer cell line, which expresses an R175H mutant p53, were purchased from Santa Cruz Biotechnology (Dallas, TX, USA). Non-small cell lung carcinoma (H1299) cells were obtained from American Type Culture Collection (ATCC; CRL-5803) and maintained in high glucose Dulbecco’s Modified Eagle Medium (DMEM) supplemented with 10% fetal bovine serum and 10 U/mL penicillin/streptomycin at 37 °C in a humidity-controlled incubator. H1299 cells are null for p53 expression; these cells were co-transfected with a tetracycline-regulated expression system plasmid (Invitrogen, ThermoFisher Scientific, Waltham, MA, USA) and pcDNA5/TO plasmid (Invitrogen, ThermoFisher Scientific, Waltham, MA, USA) containing the gene encoding R175H GOF p53 (GeneScript Biotech, Piscataway, NJ, USA). Cells expressing the R175H mutant p53 were isolated by dilution cloning and screened for the induction of homogenous p53 expression with the application of tetracycline. Selected cells were then maintained in high glucose DMEM supplemented with 10% fetal bovine serum, 10 U/mL penicillin/streptomycin, 600 μg/mL hygromycin B, 10 μg/mL blasticidin, and 1 μg/mL tetracycline at 37 °C in a humidity-controlled incubator. The treatment doses and times of FK228 (1 nM for 48 h), MS275 (5 μM for 24 h), and ZMC1 (10 mM for 72 h) were chosen based on an extensive review of the literature reporting Kd ranges for each compound and toxicity studies (e.g., [9,24,25,26,27]). Although there was some variation within the literature, we chose dosages that showed a biological effect in cell culture experiments, often choosing a mid-range dose that did not significantly affect survival. 

### 2.2. p53 Response Element Binding Assays

A p53 transcription factor assay (Cayman Chemical, Ann Arbor, MI, USA) was used to determine the DNA binding activity of p53 following the manufacturer’s instructions. Briefly, a specific double-stranded DNA sequence containing the p53 response element (RE) was immobilized onto wells of a 96-well plate and detected in a colorimetric assay. Nuclear extracts from SKBr3 cells or transfected H1299 cells were prepared following the protocol outlined by Digman and Roeder [28] or using a commercial nuclear extraction kit protocol based on this procedure (Cayman Chemical, Ann Arbor, MI, USA). A total of 10 µL of nuclear extract from untreated cells were combined with treatments (50 µM MS75 or FK228 or 1 mM DTT) and various concentrations of ZnCl_2_ and incubated overnight at 4 °C. To assure equal loading of p53 after cell treatments, a bicinchoninic acid (BCA) assay was performed on nuclear extracts isolated from the treated transfected H1299 cells followed by a sandwich enzyme-linked immunosorbent assay (ELISA) to measure the p53 levels in the nuclear extracts prior to starting the RE assay. The amount of nuclear extract added to the RE assay was adjusted based on the ELISA quantitation. Detection of R175H mutant p53 binding to the RE was determined via a spectrophotometry reading at 450/ 570 nm using a BioTek Synergy 2 Plate Reader (Agilent Technologies, Santa Clara, CA, USA). Each assay (controls vs. treatments) was performed in triplicate. 

### 2.3. Intracellular Zinc Assay 

The cell permeable FluoZin™-3 AM ester (ThermoFisher Scientific, Waltham, MA, USA) was used to determine intracellular concentrations of free zinc. DMSO-diluted FluoZin™-3 reagent was prepared according to the manufacturer’s protocol. H1299 cells expressing R175H mutant p53 were plated in 8-well chamber slides and incubated at 37 °C in a humidity-controlled incubator until 50% confluent. Cells were then treated with 1 nM FK228, 5 μM MS275, or 10 nM ZMC1 along with 5 mM of the FluoZin™-3 reagent for 15 min at 37 °C. Following incubation, each chamber was washed two times using serum free medium, then 300 μL of serum-containing medium was added, and the fluorescent intensity was measured using immunofluorescent microscopy (Leica TCS SPE Confocal, Deer Park, IL, USA). Each experiment was performed in triplicate with multiple technical replicates. 

### 2.4. ROS Detection

H1299 cells expressing R175H mutant p53 were plated at 1 × 10^4^ cells/well into either Falcon 8-well chamber slides (Corning Inc., Corning, NY, USA) or 35 mm poly-d-lysine coated, No. 1.5 coverslip bottom dishes (MatTek, Ashland, MA, USA). To measure the ROS production, cells were treated with 1 nM FK228, 5 μM MS275, or 10 nM ZMC1 for 15 min or 24 h, at which point 5 μM CellROX Green Reagent (ThermoFisher Scientific, Waltham, MA, USA) was added, and the dishes were incubated for an additional 30 min at 37 °C. Cells were washed three times using 1x PBS and fixed with 3.7% formaldehyde in 1x PBS for 15 min. After counterstaining with Hoechst 33342, cells were visualized at 630× magnification by confocal fluorescence microscopy (Leica Stellaris 5, Deer Park, IL, USA) within 24 h. Image analysis was conducted using NIH ImageJ software (version 1.52a); 10–14 fields were averaged per experiment. 

### 2.5. NanoLuc Luciferase Assay

H1299 cells expressing R175H mutant p53 were transiently transfected with the pNL (NlucP/p53-RE/Hygro) reporter vector (Promega, Madison, WI, USA) in which luciferase expression is driven by the p53 RE with TransIT^®^-LT1 transfection reagent (Mirus Bio LLC, Madison, WI, USA). Post-transfection, cells were treated with 1 nM FK228, 5 μM MS275, or 10 nM ZMC1 for 24 h. At 48 h post-transfection, cells were washed, cell stripped, counted, and diluted in Opti-MEM (Life Technologies, ThermoFisher Scientific, Waltham, MA, USA). Cells were loaded at a density of 0.025 × 10^6^ cells/well into a 96-well white plate (Costar 3917—96). A total of 25 μL of 1x Nano-Glo^®^ Live Cell Reagent (Promega, Madison, WI, USA) was added to each of the wells and the plate was gently mixed by hand for 15 s. Luminescence was measured immediately after adding the Nano-Glo^®^ live cell reagent (Promega, Madison, WI, USA) for a single time point using a signal integration time for 2 s in a 96 Microplate Luminometer (GloMax^®^, Promega, Madison, WI, USA). Four biological replicates, with three technical replicates, were performed for this experiment. 

### 2.6. Immunofluorescence Staining

All immunofluorescence experiments were performed using a standard protocol. H1299 cells transfected with R175H mutant p53 (as described above) were plated at 1 × 10^4^ cells/well (control) or 2 × 10^4^ cells/well (treated) in 8-well chamber slides. After incubation for 48–72 h, cells were fixed using 4% paraformaldehyde for 15 min, permeabilized with 0.2% Triton-X100 for 20 min, and blocked with 1% bovine serum albumin (BSA) for 1 h, followed by an overnight incubation at 4 °C with anti-p53 pAB1620 primary antibody (Santa Cruz Biotechnology, Dallas, TX, USA) at 1:50 dilution. Cells were washed three times with 1x PBS and incubated with a 1:200 dilution of Alexafluor 488 donkey anti-mouse IgG (Invitrogen, ThermoFisher Scientific, Waltham, MA, USA) for 1 h. Slides were then mounted with Fluoromount G mounting medium containing 4′,6-diamidino-2-phenylindole (DAPI, Electron Microscopy Sciences, Hatfield, PA, USA) for nuclear staining and imaged using an Axiovert Apotome microscope (Zeiss, White Plains, NY, USA) at 200x magnification with constant exposure. Image analysis was conducted using NIH ImageJ software (version 1.52a) with the ratio of fluorescent intensity to DAPI; 16 fields were averaged per experiment. 

### 2.7. p53 Immunoaffinity Purification

p53 in nuclear extracts isolated from untreated and treated H1299 cells transfected with R175H mutant p53 (Appendix A) as described above was immunoaffinity purified using agarose beads that bind to misfolded R175H mutant p53; properly folded p53 (consistent with WT p53 conformation) does not bind to the agarose beads and flows through the column, allowing for the separation of folded vs. misfolded p53 following treatments, as described above. Specifically, anti-p53 pAb240 conjugated agarose beads (Santa Cruz Biotechnology, Dallas, TX, USA) were equilibrated in wash buffer (50 mM Tris-HCl pH 7.5, 150 mM NaCl, and 1x Halt^TM^ protease inhibitors (Invitrogen, ThermoFisher Scientific, Waltham, MA, USA)) and incubated with nuclear extract for 4 h with mixing. Flow through, washes, and eluate were collected for further analysis. 

### 2.8. Immunoblotting 

To assess the nuclear concentration of p53, H1299 cells were plated at 1 × 10^5^ cells/well in a 6-well dish. When cells reached 90% confluency, protein was harvested using RIPA lysate buffer (10 mM Tris-HCL (pH 8.0), 1 mM EDTA, 1% Triton X-100, 0.1% sodium deoxycholate, and 0.1% SDS, 140 mM NaCl) with the addition of Halt^TM^ protease inhibitors plus phosphatase inhibitor (Invitrogen, ThermoFisher Scientific, Waltham, MA, USA). Nuclear lysates were isolated as described previously and protein concentrations were determined using a BCA assay or a NanodropTM ND-1000 Spectrophotometer (ThermoFisher Scientific, Waltham, MA, USA). A total of 30 μg of protein from each sample was loaded into a pre-cast 4–20% SDS-PAGE gel (BioRad, Hercules, CA, USA) and electrophoretically separated at 130 volts for 75 min and transferred to a polyvinylidene fluoride (PVDF) membrane following standard protocol. The membrane was incubated with a 1:1000 dilution of anti-p53 pAb240 (Santa Cruz Biotechnology, Dallas, TX, USA) and anti-p53 primary antibodies (Cell Signaling Biotechnology, Dallas, TX, USA); a 1:1000 dilution of anti-GAPDH primary antibodies was used as a loading control. Following primary antibody incubation, the membrane was incubated with a 1:5000 dilution of AlexaFluor-700 and AlexaFluor-800 secondary antibodies (Abcam, Waltham, MA, USA). Relative protein expression was quantified using an Odyssey CLx Imaging system (Li-Cor Biosciences, Lincoln, NE, USA), normalized to glyceraldehyde 3-phosphate dehydrogenase (GAPDH). For the assessment of post-translational modifications, Immunoblotting was performed as described above with the use of 1:100 dilutions of anti-phospho-p53 (Ser15), anti-phospho-p53 (Ser315), and anti-acetyl-p53 (Lys382) primary antibodies (Cell Signaling Biotechnology, Danvers, MA, USA).

### 2.9. ELISA

The proportion of misfolded to folded R175H p53 mutant protein following treatment was determined using ELISA. R175H p53-transfected H1299 cells were plated in 6-well dishes at 2 × 10^5^ cells/well and incubated with either 1 nM FK228 or 5 μM MS275 for 48 h, or with 10 nM ZMC1 for 72 h. Untreated cells were used as a control. Nuclear extracts were harvested from the control and treated cells using the Nuclear Extraction Kit (Cayman Chemical, Ann Arbor, MI, USA) following the manufacturer’s instructions as described above. A total of 10 μL of each nuclear extract was added to a 96-well plate and a 1:500 dilution of anti-p53 pAb240 primary antibody (Santa Cruz Biotechnology, Dallas, TX, USA) was added and incubated overnight at 4 °C. After washing, a 1:1000 dilution of an HRP-conjugated secondary antibody was added and the optical density was measured at 450 nm using a BioTek Synergy 2 plate reader (Agilent Technologies, Santa Clara, CA, USA). Each assay was performed in triplicate. 

### 2.10. Statistical Analysis

Statistical significance was determined using a Dunnett’s multiple comparisons test (control vs. treatment). For comparisons among different treatments, a one-way ANOVA Tukey multiple comparisons test was used with an alpha value set to 0.05. All data were analyzed and graphed using GraphPad Prism version 9.0 (GraphPad, La Jolla, CA, USA).

## 3. Results

### 3.1. HDACi with Dissimilar ZBGs Differentially Impact R175H Bind to RE In Vitro

Although the refolding of the p53 R175H GOF conformational mutant in response to elevated zinc is well-established, much of this work has been conducted with purified protein. Cellular zinc is in a complex equilibrium, with free zinc (Zn^+2^) maintained at 5–6 orders of magnitude lower than bound or sequestered Zn^+2^. To examine the ability of Zn^+2^ to refold and reactivate R175H in the context of a cellular pool of bound zinc, an in vitro p53 RE binding assay was used. Specifically, nuclear extract from the SKBr3 human breast cancer cell line expressing R175H was combined with increasing amounts of ZnCl_2_ and the resulting levels of binding to the p53 RE binding were assayed. Increasing levels of ZnCl_2_ led to significantly increased RE binding at all ZnCl_2_ concentrations at or above 25 μM (Figure 1A).

The ability of the ZBG to shift R175H p53 RE binding was then assayed using a constant concentration of 50 µM ZnCl_2_. We observed a significant increase in the R175H p53 RE binding only with the addition of 50 μM MS275; the addition of 50 µM FK228 or 50 µM ZMC1 did not change the amount of p53 RE binding compared to the control (Figure 1B). We repeated the FK228 assay with and without additional exogenous ZnCl_2_ (Figure 1C). Interestingly, without the addition of exogenous ZnCl_2_, treatment with 50 µM K228 reduced p53 RE binding. In addition, when FK228 was reduced by the addition of 1 mM DTT, FK228 significantly decreased p53 RE binding in the presence of 50 µM ZnCl_2_, an effect that was abolished with the addition of 100 µM ZnCl_2_ (Figure 1C). Although the p53 RE binding following treatment with MS275 was higher in the presence of ZnCl_2_ (Figure 1B,D), it remained significantly higher in the absence of exogenous ZnCl_2_ (Figure 1D). Controls included a WT p53 positive control with and without the addition of a competitor dsDNA sequence as a specificity control (provided by the manufacturer; Appendix A) and the zinc metallochaperone ZMC1, which showed no effect on p53 RE binding at 50 µM compared with the control (Figure 1B). 

### 3.2. MS275 Benzamide ZBG Increased Intracellular Free Zinc and ROS Production 

To directly assess the ability of HDACi ZBGs to impact intracellular zinc, we assayed free Zn^+2^ 15 min after adding HDACi to the cell cultures. Treatment with 5 µM MS275 increased the intracellular free Zn^+2^ over the untreated cells but the levels were not statistically different to that of the zinc metallochaperone ZMC1 at 10 nM (Figure 2A). As expected, due to the ~8 nM Kd of FluoZin^TM^-3 for free Zn^+2^, no significant change in zinc concentration was observed with the 1 nM FK228 treatment. Detection of ROS downstream of elevated Zn^+2^ levels was complicated by the fact that p53 activation affects the ROS levels [29]. Therefore, we measured the ROS production at two time points after HDACi treatment: 45 min and 24 h. The 45-minute time point was included as changes in HDACi-induced gene expression affecting ROS production were expected to be minimal, allowing for a comparison of ROS production before and after any treatment-induced changes in gene expression. At both time points, we observed a significant increase in ROS production following treatment with both ZMC1 and MS275, but not with FK228, compared to the control (Figure 2B,C). Of note, the degree of ROS production in cells treated for 24 h with ZMC1 was significantly higher than for cells treated with MS275 (Figure 2C).

### 3.3. Treatment with the HDACi MS275 Refolds and Reactivates R175H p53

While the p53 RE binding assay (Figure 1) supports Zn^+2^ and MS275 reactivating R175H mutant p53 to improve binding to the p53 RE, the doses used are non-physiological levels of Zn^+2^ and HDACi. Therefore, we used a cell-line model to address refolding and reactivation of R175H p53 at more physiologically relevant levels of HDACi. As the p53-null H1299 non-small cell lung carcinoma cell line is an established p53-null system to study mutant p53 [30,31], we expressed R175H mutant p53 in a tetracycline inducible system to reproduce the previously reported [30] increase in cell proliferation with R175H mutant p53 expression and the expected decrease following ZMC1 and HDACi treatment (Appendix A). However, as multiple mechanisms have been documented for the decreased proliferation downstream of HDACi treatment, we next focused specifically on the refolding and reactivation of the R175H GOF p53 mutant protein in this model. 

To determine the effects of treatments on the R175H mutant p53 structure, we utilized characterized monoclonal antibodies that bind to different p53 epitopes; the pAb1620 antibody binds to an epitope formed with the coordination of zinc, allowing for the proper folding of p53 while pAb240 binds to an epitope that is only exposed when zinc is not bound, suggestive of a misfolded p53. Specifically, appropriate coordination of zinc folds the p53 protein, bringing amino acids 145–157 adjacent to amino acids 201–212 to create the epitope bound by the monoclonal antibody pAb1620 [32,33,34]. In contrast, the epitope bound by pAb240 consists of amino acids 211–217, which are only accessible when the p53 protein does not coordinate zinc or is otherwise misfolded [33,35].

The amount of R175H mutant p53 that has refolded following treatment with HDACi or ZMC1 was assessed by immunofluorescent staining using the pAb1620 antibody. Cells treated with either 10 nM ZMC1 or 5 μM MS275 generated a strong fluorescent signal, suggesting that these treatments resulted in refolding of the R175H mutant p53 expressed in these cells (Figure 3A, middle), which was significantly higher than the untreated cells (Figure 3B). Interestingly, little to no fluorescence was detected in the cells treated with 1 nM FK228 compared to the untreated cells (Figure 3A,B), suggesting that FK228 does not refold R175H mutant p53, at least not in the same way as the other treatments. While the conformational antibody data suggest the refolding of R175H mutant p53, the data are not indicative of reactivation. To assess this, we utilized a p53 RE luciferase reporter construct. Interestingly, treatment with both HDACi showed significantly increased p53 RE binding as measured by luciferase expression (Figure 3C), suggesting that FK228 treatment leads to increased RE binding without the coordination of Zn^+2^. As both HDACi increased luciferase expression more than ZMC1, our data suggest the possibility that additional factors, independent of the zinc-mediated refolding of p53, affect the levels of p53 RE binding in vivo. 

We next assessed R175 mutant p53 RE element binding using nuclear extracts from the treated cells. As the expression of R175H mutant p53 varies substantially with treatments (Appendix A), care was taken to normalize for p53 expression. First, we adjusted the volume of nuclear extract added to the p53 RE binding assay using immunoblot data. Second, the p53 levels in this volume of extract was measured by ELISA and the signal obtained in the p53 RE binding assay was then normalized to the total amount of p53. Our results showed that all treatments increased R175H mutant p53 RE binding compared to R175H mutant p53 from the untreated cells (control); extracts from H1299 cells, null for p53 expression, were used as a negative control (Figure 4A). Interestingly, when these conditions were directly compared, nuclear extracts from the cells treated with MS275 showed significantly greater p53 RE binding activity compared to ZMC1 (*p* < 0.05, not indicated on the graph).

To assess whether treatments resulted in a stable refolding of R175H mutant p53, we performed immunoaffinity purification and an ELISA using the p53 conformational antibody pAb240 that binds only to the misfolded p53 protein [30,33]. Specifically, misfolded R175H mutant p53 was immunoaffinity purified using commercially available agarose beads and the bound (misfolded) and unbound (folded) fractions were analyzed by immunoblot; visually, treatment with both ZMC1 and MS275 resulted in less misfolded p53 than treatment with FK228 (Figure 4B and Appendix A). Consistent with recent models of p53 folding [23], analysis of WT p53 in MCF7 nuclear extracts yielded ~25% misfolded p53. Therefore, to improve sensitivity, we also performed an ELISA using an anti-p53 pAb240 antibody to determine the amount of misfolded p53 in nuclear extracts from cells treated with ZMC1 and MS275; the results obtained support our immunoblot data, indicating a significant reduction in misfolded p53 (Figure 4C). 

### 3.4. Activating Post-translational Modifications (PTMs) on R175H with HDACi Treatment

The activity of WT p53 is modulated by a myriad of post-translational modifications (PTMs) ([36], reviewed in [37]). Activation of p53 by DNA damage increases phosphorylation on serine 15 and 315 ([30,38,39] and Appendix A). In addition, acetylation on the C-terminal lysine residues precedes DNA binding (reviewed in [40]) and HDACi have been shown to alter the acetylation levels of p53 [41]. Therefore, we next addressed how treatments affected the PTMs on R175H mutant p53. Using anti-acetyl p53 and anti-phospho p53 antibodies, our data showed that R175H mutant p53 in the untreated (control) and treated cells appeared to be acetylated on lysine 382 and phosphorylated on both serine 15 and serine 315, as detected by immunoblot (Figure 5A and Appendix A). Interestingly, the anti-phospho-Ser315 antibody detected multiple bands at higher molecular weights, potentially consistent with p53 dimers and tetramers (Appendix A), which are also present in lysates from UV exposed cells (Appendix A). In addition, a 35 kDa immunoreactive band was present in some samples ([42], Appendix A). A side-by-side comparison of PTMs is complicated as the expression levels of total p53 changes with treatment (Appendix A), and the accumulation of high levels of the R175H mutant p53 protein occurs, which is characteristic of some p53 mutant proteins. To remove cytosolic p53 from the analysis, treated cells were fractionated, and phosphorylation on serine 15 in the nuclear fraction was quantitated by immunoblotting. After treatment with 10 nM ZMC1, we observed a statistically significant increase in R175H mutant p53 phosphorylated on serine 15 when normalized against total p53 in the nuclear fraction (Figure 5B).

## 4. Discussion

In this work, we set out to test whether the zinc binding groups (ZBG) common to HDACi have the capacity to alter intracellular zinc concentrations and affect the activity of other zinc-requiring enzymes. As a class, HDACi inhibit their target enzymes through Zn^+2^ binding and are potent anti-neoplastic drugs. Both entinostat (MS275) and romidepsin (FK228) have been successfully used, often in combination with other therapies, in the treatment of some types of cancer including breast cancer [43,44], colon cancer [45], endometrial cancer [46], and T cell lymphoma [47,48,49]. However, the therapeutic potential of HDACi has been limited due to their many off-target effects (as recently reviewed in [1]). While in vitro studies define the specificity of HDACi for specific class I HDAC enzymes, studies suggest that treatment with HDACi results in broad-ranging effects and leads to substantial changes in approximately one third of the genome. Clearly, a major contribution to this lack of specificity is that the inhibition of HDACi leads to alterations in the acetylation state of many signaling proteins including p53 [24,41], in addition to changing the overall structure of chromatin by altering histone acetylation. This work identified alterations in free Zn^+2^ as a potential additional factor that contributes to the broad range of HDACi activities.

In vitro data presented here substantiate a role for alterations in the Zn^+2^ levels in the downstream effects of HDACi and that these effects may vary by nature of the ZBG. As cellular free Zn^+2^ levels are kept in the picomolar range [10,11,12], the ZBG is a sensitive target. Utilizing HDACi with two different ZBGs suggests that Zn^+2^ may either be effectively increased or decreased. As FK228 only binds Zn^+2^ once it is reduced intracellularly, it theoretically has the capacity to reduce free Zn^+2^ concentrations. In contrast, MS275 is expected to bind Zn^+2^ at high extracellular concentrations, and upon cellular uptake, bring Zn^+2^ into the cell. Our results suggest that while the effects of FK228 appear to be solely related to zinc, in that effects are abolished by the addition of ZnCl_2_, a direct interaction between MS275 and R175H mutant p53 is suggested by our in vitro p53 response element binding data that implies that the cap and linker regions allow interaction with multiple zinc finger proteins including R175H mutant p53, a finding that is consistent with the observation that the cap and linker groups influence the activity of the ZBG of different HDACi [50]. Although ZMC1 binds Zn^+2^, we did not detect any measurable effect on R175H p53 binding to its response element following the addition of Zn^+2^, a finding consistent with its role as an ionophore [17,25].

Interpretation of the cellular data is more complex. The measurement of intracellular free Zn^+2^ concentrations is limited by the available fluorescent indicators and is complicated by the equilibria between free Zn^+2^, the Kd of the zinc indicator, and the pool of bound zinc. Although the observed elevation in zinc after MS275 treatment is not significantly different than that observed with ZMC1, this is consistent with the hypothesis that MS275 binds extracellular Zn^+2^ to bring zinc into the cell. Although not directly addressed here, the known ability of ZMC1 to bind copper at a higher affinity than zinc may have an important role in the toxicity of ZMC1 relative to MS275 in other settings. Overall, our data have implications for the development of combination therapies targeting zinc [19,26].

Data generated after FK228 treatment of cells are more nuanced. Reduced FK228 shows the potential for reducing Zn^+2^ concentration, as our results demonstrated that the addition of DTT reduced p53 binding in an in vitro p53 RE binding assay. Immunostaining with the pAb1620 conformational antibody suggests a notable absence of zinc coordination after the treatment of cells with FK228, consistent with a decrease in free Zn^+2^ downstream of FK228 treatment. However, given the ~8 nM Kd of FluoZin-3 for free Zn^+2^, a decrease in free Zn^+2^ concentration downstream of 1 nM FK228 was not detectable using this or other available fluorescent indicators of free Zn^+2^. The increase in detectable ROS downstream of FK228 treatment may reflect a difference in the sensitivity of these two assays and, if zinc-related, reflective of disturbance of the complex equilibrium between bound and free Zn^+2^ after FK228 treatment. However, the activity of R175H mutant p53 in RE binding assays both in vitro and in cells suggests that the activity may be independent of zinc, potentially downstream of activating PTMs. Multi-step models for the reactivation of p53 have been previously proposed for both the isolated DNA binding domain of p53 and the intact protein [23,51]. With PTMs leading to conformational changes in p53, the reactivation of R175H mutant p53 without zinc coordination may be consistent with other published studies.

While the ability of HDACi to modulate zinc bioavailability in cell extracts and intracellularly is supported by the work presented here, more studies are needed to explore the downstream consequences of elevations in zinc. As revised estimates determine the FluoZin3 Kd for free Zn^+2^ at ~8 nM Kd [52] and Zn^+2^ is typically maintained in the picomolar range, the detected levels are likely to represent ≥10-fold elevation in free Zn^+2^ concentrations. Although the observed elevation in Zn^+2^ appears to be transitory, our data suggest that zinc appears to have significant downstream effects in both the elevation in ROS levels and in the reactivation of the R175H mutant p53 protein.

The ROS production suggests that treatment with ZMC1 is more toxic to the cells, as measured solely by intracellular ROS, than treatment with HDACi. However, both ZMC1 and MS275 resulted in a significant increase in ROS production within 45 min, compared to the untreated cells or cells treated with FK228. Interestingly, there was significantly higher ROS production following 24-h treatment with ZMC1 than following MS275 treatment, while 24-h treatment with FK228 yielded no greater ROS production than in the untreated cells. These data are in agreement with several other studies linking ROS and HDACi [53,54,55]. Of notable importance, MS275 has been shown to increase ROS [26], a link that has been replicated with other HDACi [56]. However, HDACi have also been shown to reduce the transcription of ROS-producing NADPH oxidase [57], and ROS production (at least in the mitochondria) has been linked to the inhibition of non-nuclear HDAC enzymes [58]. Adding to this complicated picture is that the activity of p53 itself modulates ROS [29]. The data presented here suggests that FK228 may negatively affect the activity of ROS-producing enzymes more so than MS275, but additional experiments are needed to make this determination.

When placing these data into context, several important points need to be considered. First, all in vitro experiments were performed with nuclear extracts and not purified R175H mutant p53 protein. Thus, after determining that all concentrations at or above 25 µM ZnCl_2_ led to significant increases in R175H mutant p53 binding to the p53 RE, we chose the 50 µM dose to provide the dynamic range necessary to detect both increased and decreased RE binding. Although the experimentally determined value of 50 μM ZnCl_2_ was high when compared to experiments utilizing purified protein, the presence of many zinc-binding proteins in the nuclear extracts was expected to buffer free zinc. Free Zn^+2^ in the sample was not directly measured and only assessed using R175H mutant p53 protein refolding. As our focus was the interaction of HDACi with ZnCl_2_, treatments were added at a 1:1 molar ratio. Thus, like many in vitro assays, the amount of added inhibitor was substantially higher than that used to treat cells [14]. A consequence of this experimental approach is that the results presented here are not directly comparable to the published characterization of the zinc dependence of mutant p53 refolding [23].

A strength of the in vitro approach is that cellular responses to either the reactivation of the R175H mutant protein or changes in gene expression due to HDACi treatment do not complicate data interpretation. These issues are unavoidable when working in cell lines. To minimize these impacts, we initially focused on timepoints where changes in gene expression are not a complicating factor. Specifically, free Zn^+2^ was measured after 15 min of treatment. While zinc homeostatic responses are expected to make any increase in zinc transient, we detected a significant increase in intracellular zinc at this early timepoint. The 8 nM kD of the fluorescent probe suggests that this increase is well above the levels of free zinc described in the literature. ROS production, potentially downstream of increased zinc, was measured after 45 min of treatment including 30 min of activation of the fluorescent ROS probe. This timeframe is well before changes in gene expression downstream of HDACi treatment and/or R175H mutant p53 reactivation are expected to occur. As these complicating factors cannot be avoided when attempting to interpret data taken at later timepoints, these data were included to suggest that variability in zinc and ROS levels downstream of the HDACi may contribute to differential activation of R175H mutant p53. Many additional experiments will be needed to definitively establish differences in R175H mutant p53 protein activation separate from the effects of HDACi.

While transient increases in Zn^+2^ are sufficient to reactivate R175H mutant p53 [15], other zinc finger proteins may not be equally affected. p53 activity appears to be unusually sensitive to variations in zinc concentrations [23], with other zinc finger proteins having significantly lower Kds in the picomolar range [59]. It has been hypothesized that the sensitivity of p53 to Zn^+2^ concentration plays a regulatory role [23], a level of regulation that may extend to other proteins [10,11,12]. However, the observed reactivation of the R175H conformational mutant p53 protein supports the postulate that the ZBGs of HDACi may affect the zinc levels enough to affect the activity of at least a subset of cellular zinc-binding proteins. These findings suggest that zinc may play an unusually important role in the regulation of p53 activity as p53 conformation appears to be exceptionally dependent on Zn^+2^. Thus, the ability of HDACi to alter zinc bioavailability may have profound consequences for cancers bearing p53 mutations. Taken together, our results demonstrate the variable effects different HDACi have on p53 activity, intracellular zinc levels, and cellular health that should be considered during the development of novel, or combinatorial, epigenetic therapies for use in cancer treatment.

## 5. Conclusions

The results presented here suggest that an additional contribution to the observed dramatic changes in gene expression may be alterations in zinc levels downstream of HDACi treatments. Altered zinc levels have the potential to impact a wide variety of zinc finger proteins in transcription factors and other nuclear proteins, and this may also contribute to the wide-ranging biological effects of HDACi. Future studies focusing on how HDACi modulate zinc bioavailability and free radical production may aid in the development of more specific HDACi to be used as therapeutic agents, alone or in combination with existing therapies, to prolong patient survival and potentially reduce off-target adverse effects.

## Figures and Tables

**Figure 1 biomolecules-13-01588-f001:**
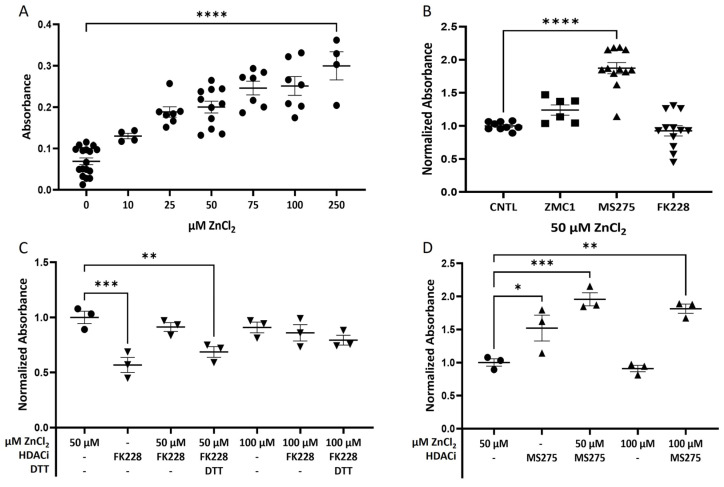
In vitro measurement of R175H mutant p53 response element (RE) binding. (**A**) R175H mutant p53 binding to a p53 RE increases with an increasing concentration of exogenous ZnCl_2_ at or above 25 μM. (**B**) In 50 μM exogenous ZnCl_2_, the addition of 50 μM of the HDACi MS275 significantly increased p53 RE binding, with no increase observed following the addition of 50 μM of the HDACi FK228 or 50 μM of the zinc metallochaperone ZMC1. (**C**) At 50 μM exogenous ZnCl_2_, the addition of 50 μM FK228 did not increase the p53 RE binding, but both the absence of zinc or the addition of DDT resulted in a decrease in p53 RE binding that was abolished when a higher concentration of ZnCl_2_ (100 μM) was used. (**D**) In the absence of exogenous ZnCl_2_, 50 μM MS275 increased p53 RE binding over the control with the addition of 50 or 100 μM ZnCl_2_, resulting in significantly higher p53 RE binding. Horizontal lines with error bars represent the mean ± SEM. * denotes *p* < 0.05, ** denotes *p* < 0.01, *** denotes *p* < 0.001, **** denotes *p* < 0.0001.

**Figure 2 biomolecules-13-01588-f002:**
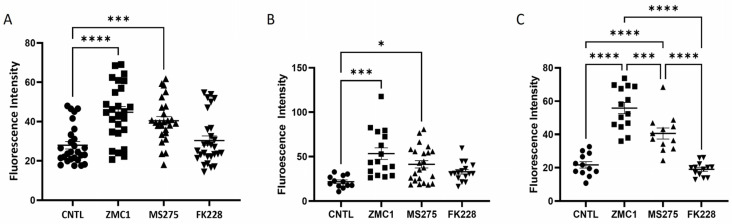
(**A**) FluoZin^TM^-3 assay measuring intracellular concentrations of free Zn^+2^ following treatment with 10 nM ZMC1, 5 μM MS275, or 1 nM FK228 for 15 min. Compared to untreated cells (control), both ZMC1 and MS275 significantly increased the intracellular concentration of free Zn^+2^ while FK228 had no effect. (**B**,**C**) CellROX assay measuring intracellular ROS production following treatment of cells for 45 min (**B**) or 24 h (**C**). Compared to the untreated cells (control, CNTL), ROS production was significantly elevated in the cells treated for either 45 min or 24 h with ZMC1 and MS275, with significantly higher ROS production following ZMC1 treatment compared to MS275 after 24 h. No increase in ROS production was observed when the cells were treated for either 45 min or 24 h with FK228. Horizontal lines with error bars represent the mean ± SEM. * denotes *p* < 0.05, *** denotes *p* < 0.001, **** denotes *p* < 0.0001.

**Figure 3 biomolecules-13-01588-f003:**
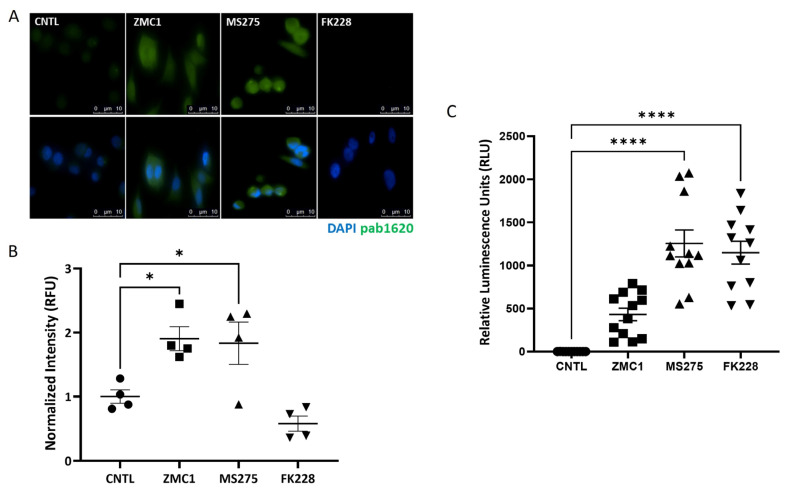
In vivo refolding and reactivation of R175H mutant p53. (**A**) Cells treated with either 10 nM ZMC1 or 5 μM MS275, but not 1 nM FK228, resulted in the refolding of R175H p53, as identified by anti-p53 pAb1620 binding (green). Nuclei are stained with DAPI, as indicated in the bottom four panels. (**B**) Quantification of fluorescent intensity, normalized to DAPI, show that ZMC1 and MS275 treatment resulted in a significantly increased fluorescent intensity compared to the control. Treatment with 1 nM FK228 had no effect on the refolding of R175H p53. (**C**) An increase in p53 RE binding, as measured by luciferase activity, was observed when the cells were treated with either 1 nM FK228 or 5 μM MS275, but not with 10 nM ZMC1. Horizontal lines with error bars represent the mean ± SEM. * denotes *p* < 0.05, **** denotes *p* < 0.0001.

**Figure 4 biomolecules-13-01588-f004:**
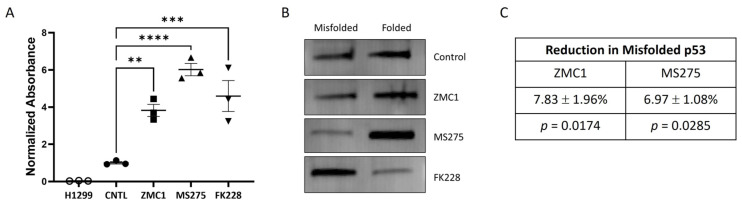
Measurement of R175H mutant p53 RE binding following treatment. (**A**) Compared to the untreated H1299 cells transfected with R175H mutant p53 (control), cells treated with either 10 nM ZMC1, 5 μM MS275, or 1 nM FK228 showed a significant increase in their R175H mutant p53 RE binding. Untransfected H1299 cells, null for p53 expression, served as a negative control. Horizontal lines with error bars represent the mean ± SEM. ** denotes *p* < 0.01, *** denotes *p* < 0.001, **** denotes *p* < 0.0001. (**B**) Immunoblotting of purified nuclear extracts from the treated and untreated cells showed that treatment with ZMC1 or MS275 resulted in a greater proportion of p53 that was in the folded vs. the misfolded configuration, while treatment with FK228 showed the opposite pattern. Immunoblot original image can be found in Appendix A. (**C**) An ELISA using an anti-p53 pAb240 antibody performed on nuclear extracts isolated from cells treated with ZMC1 or MS275 confirmed that treatment resulted in a significant reduction in the fraction of misfolded R175H mutant p53.

**Figure 5 biomolecules-13-01588-f005:**
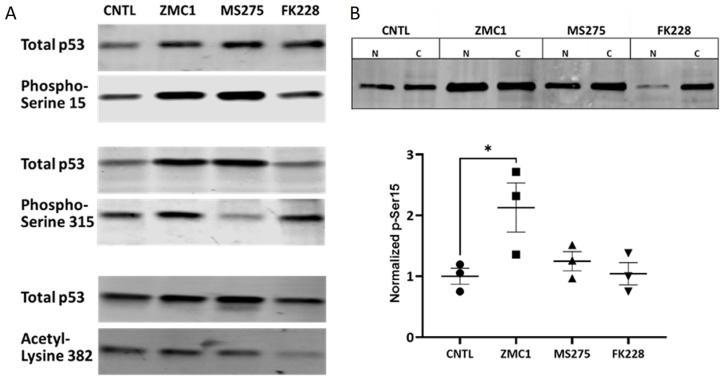
Detection of post-translational modifications (PTMs) present on R175H mutant p53 following treatment. (**A**) Immunoblot using anti-phospho-Ser15, anti-phospho-Ser315 p53, and anti-Lys382 antibodies on the cell lysates obtained from the untreated cells vs. cells treated with either 10 nM ZMC1, 5 μM MS275, or 1 nM FK228. Results showed that all PTMs were present on R175H mutant p53 but in varying abundances. Immunoblot original images can be found in Appendix A. (**B**) Immunoblot (top) and quantification (bottom) of p53 phosphorylation on serine 15 in nuclear (N) and cytosolic (C) extracts, normalized to total p53 by immunoblotting revealed a significant increase in this PTM in nuclear lysates from cells treated with 10 nM ZMC1 (* *p* < 0.05). Immunoblot original image can be found in Appendix A.

## Data Availability

The data presented in this study are available on request from the corresponding author.

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
