# Peer review of "Biological Effects of HDAC Inhibitors Vary with Zinc Binding Group: Differential Effects on Zinc Bioavailability, ROS Production, and R175H p53 Mutant Protein Reactivation"

_biomolecules, 2023, doi:10.3390/biom13111588_

Round 1

Reviewer 1 Report

Comments and Suggestions for Authors

This manuscript discusses the biological effects of HDAC inhibitors with two different zinc-binding groups. The authors claim that the benzamide inhibitor MS275 can bring

Zn2+ into the cell, thereby increasing intracellular Zn2+ levels and inducing high levels of ROS. However, the data presented are in part contradictory and cannot exclude additional cellular pathways that could lead to similar results. Furthermore, the BCA assay to determine R175H binding to RE was performed in the presence of HDACi carrying the zinc binding group, which act as chelator, although the BCA assay is incompatible with chelators. Therefore, the data obtained in Figure 1 are questionable. Furthermore, R175H binding to RE is only observed at >25 µM Zn2+, which is 6 orders of magnitude! above the free Zn2+ levels in cells. This raises doubts that R175H-RE binding could be used to indicate intracellular Zn2+ concentrations. In addition, the data shown in Figure 2 are supposed to show an increase in intracellular free Zn2+ and ROS production upon addition of MS275 and FK228. In my opinion, the highly variable and overlapping data are not convincing to support this statement. In this experiment, no higher concentrations of Zn2+ are added to the cells, which could possibly be transported across the cell membrane. Therefore, only a small amount of Zn2+ would possibly be transported into the cell after treatment with the compound.

On the other hand, the cell has very efficient mechanisms to maintain the intracellular Zn2+ concentration. Zn2+ is buffered to pM concentration by protein binding sites.

For example, in cultured HT29 colon cancer cells, about 10% of high affinity sites were found to be demetallated, contributing to a buffer ratio of ~10/90% under these conditions (doi:10.1016/j.jinorgbio.2003.10.010). An additional mechanism to maintain Zn2+ levels is the rapid transport of excess zinc ions into a subcellular store (muffling), which is not visible in Figure 3A. Considering the thermodynamic Zn2+ buffering and the kinetic muffling, it is difficult to understand, why the Zn2+ concentration should increase in the chosen experimental setup. 

Furthermore, the cellular data on the effects of the compounds on Zn2+ levels and ROS are contradictory. The data in 3A may indicate that FK228 does not refold R175H, but increases RE binding, which was expected to occur only with folded R175H. The authors interpret this contradiction to mean that "additional factors, independent of zinc, affect the level of p53 RE binding in vivo".  But why should such additional and unidentified factors only play a role in the presence of FK228?

The problem with the data presented in general is that they focus on clear expected effects and neglect possible bypass or alternative cellular processes, such as epigenetic up- and down-regulation of different genes, or non-specific interactions with (unknown) off-targets, which would produce similar results. And there is no doubt that the compounds have additional effects on the cells, since the treatment with them affects their proliferation.

The authors say that Zn2+ would increase ROS levels and that Zn2+ levels are increased by MS275. However, this could be a coincidence rather than a causal relationship.

There is clear evidence that another Zn2+ chelator, SAHA, inhibited HeLa cell growth via caspase-dependent apoptosis, which was influenced by mitochondrial O2 and Trx1 levels (https://doi.org/10.3892/ijo.2014.2337). However, the increase in ROS levels was exclusively produced via this SAHA-induced caspase pathway. A Zn2+ mediated increase in ROS levels could be excluded in this study.

Therefore, I would recommend that this study not be published because the data presented, which are contradictory and highly scattered, do not provide sufficient evidence for the main conclusion that HDAC inhibitors cause significant changes in zinc levels leading to an increase in ROS levels or other biological effects not mentioned. 

Minor issure:

-Write "Zn2+"  or "zinc ion" instead of "Zinc", which is a solid metal.

-"..despite the many obstacles encountered in moving ... into the clinic."    Please explain, which obstacles are meant.

-"...HDACi alter the acetylation of histones..." That is not true. Selective HDAC6 inhibitors, for example, have no impact on histone acetylation.

-"..HDAC inhibition leads to ... on p53." This statement is too general. Please be more precise and discuss such effects in context with

the specific isozyme, which is involved.

-Fig. 1: What does "Normalized Absorbance" mean exactly? How much is the absolute absorbance? A value above 2 is certainly unreliable.

The values for 50 µM Zinc in 1A and the same experiment CNTL (Control?) in Fig. 1B are not comparable.

x-axis: Write Zn2+ instead of Zinc, which is a solid metal.

-The effects of different concentrations of MS275, FK228 and ZMC1 were compared. What was the rationale for selecting the concentrations used?

Fig. S1:

Please add charges to Zn 

Reviewer 2 Report

Comments and Suggestions for Authors

The manuscript by Flores et. al. reports the effects of HDAC inhibitors having Zinc binding groups on zinc bioavailability and its potential toxicity. The manuscript is well written, and the study is well designed with thorough results and a discussion section. I support the publication of this manuscript after one minor change.

·         It is  important to mention what form of zinc was used in this study. Was it elemental Zn or Zn ion as chloride/acetate salt or any other form? Especially in Figure 1 all panels and in methods.

Reviewer 3 Report

Comments and Suggestions for Authors

In my opinion, the work is highly relevant and suitable for Biomolecules MDPI. The scientific writing is good, while the experimental design is well performed. Significant novel findings on the implication of HDAC inhibitors on zinc concentration, ROS levels, and activity of R175H mutant p53 protein. Data are robust and adequately discussed. Here are some minor suggestions for improving the work.

a.       The main observation is that authors should clearly state the clinical utility and future applications of the study findings

b.       Please include study limitations before conclusions

c.       Abbreviations such as ZBG should be explained the first time being mentioned. This would improve the readability of the work for non-expert readers. Please also check the work for other abbreviations.

d.       I suggest including more supporting references in the first part of the introduction.

e.       Although the non-specific activities of HDAC inhibitors, they are used in the clinic for antitumor purposes (PMID: 35350569, PMID: 31816600). This information should be included  

f.        For a better reading I discourage mentioning figures in the discussion

g.       Concerning the modulation of ROS levels and HDAC inhibitors, additional works have been published PMID: 27498117, PMID: 18852132 and

Round 2

Reviewer 1 Report

Comments and Suggestions for Authors

The authors have amended the manuscript and have clarified some aspects, such as the RE binding assay.

However, intracellular buffering is an extremely fast process, and it is not comprehensible how substances in the micromolar range can cause a measurable change in the very low pM zinc concentration. Despite the statistical analysis, the quality of the data is very low and still does not convince me that the substances in the presence of high intracellular protein concentrations have an influence on the strongly lowered zinc concentration due to (very fast) buffering. Also, alternative processes, which lead to the observed effects cannot be excluded. 

Therefore, I stand by my recommendation not to publish the paper at biomolecules.

Author Response

This manuscript is not meant to exclude alternative processes and future experiments are planned to delineate how the ZBG affects HDACi function. However, we feel that the data, utilizing both nuclear extracts and cell-based response element binding assays, is sufficient to indicate that the ZBG of HDACi may contribute to the complexity of cellular responses to HDACi treatment. We look forward to addressing other concerns in future work.